# Computational Analysis and Experimental Testing of the Molecular Mode of Action of Gatastatin and Its Derivatives

**DOI:** 10.3390/cancers15061714

**Published:** 2023-03-10

**Authors:** Paola Vottero, Qian Wang, Marek Michalak, Maral Aminpour, Jack Adam Tuszynski

**Affiliations:** 1Department of Biomedical Engineering, University of Alberta, Edmonton, AB T6G 2V2, Canada; 2Department of Biochemistry, University of Alberta, Edmonton, AB T6G 2H7, Canada; 3Department of Physics, University of Alberta, Edmonton, AB T6G 2E9, Canada; 4Dipartimento di Ingegneria Meccanica e Aerospaziale (DIMEAS), Politecnico di Torino, I-10129 Turin, Italy

**Keywords:** tubulin, microtubules, gatastatin, docking, molecular dynamics

## Abstract

**Simple Summary:**

The glaziovianin A derivative gatastatin, presented as a γ-tubulin-specific inhibitor, could represent a viable chemotherapeutic strategy to solve the specificity issues associated with targeting α and β tubulin. Since gatastatin’s specificity for γ tubulin has not been confirmed by an in silico analysis or verified experimentally by other groups, we undertook finding a molecular-level elucidation of the binding mode of gatastatin and comparing its predicted binding affinity values for both α-β and γ tubulin. We believe that our paper opens the possibility for the rational design of a long-sought candidate drug with desired specificity and selectivity for γ tubulin.

**Abstract:**

Given its critical role in cell mitosis, the tubulin γ chain represents a viable chemotherapeutic target to solve the specificity issues associated with targeting α and β tubulin. Since γ tubulin is overexpressed in glioblastoma multiforme (GBM) and some breast lesions, the glaziovianin A derivative gatastatin, presented as a γ-tubulin-specific inhibitor, could yield a successful therapeutic strategy. The present work aims to identify the binding sites and modes of gatastatin and its derivatives through molecular-docking simulations. Computational binding free energy predictions were compared to experimental microscale thermophoresis assay results. The computational simulations did not reveal a strong preference toward γ tubulin, suggesting that further derivatization may be needed to increase its specificity.

## 1. Introduction

### 1.1. Microtubule Polymerization and Nucleation

Microtubules (MTs) represent major cytoskeletal filaments in all eukaryotic cells. They emanate from the centrosome, near the cell nucleus, and develop as a cytoplasmic network up to the cortical region. As interconnecting structures, they are involved in many cellular processes, such as motility, signal sensing, cell organization, structural strength, intracellular transport, and chromosome segregation during mitosis.

Microtubule polymerization is regulated by a molecule called guanosine triphosphate (GTP). GTP is a purine nucleoside triphosphate exerting various functions in the cell, from being an energy source for protein synthesis and gluconeogenesis to acting as a signal transducer. Concerning microtubules, GTP is responsible for the mechanism known as dynamic instability, i.e., the alternation of assembly and disassembly cycles.

The α-β heterodimer hosts a GTP molecule at the dimerization interface in a nonexchangeable site. Upon the formation of the dimer, another GTP molecule binds to the exposed β subunit in the exchangeable site located at the interdimer interface, as shown in Figure 1. This second GTP molecule, being more exposed to solvents, can be hydrolyzed to GDP during or shortly after polymerization; GTP hydrolysis is thought to provide MTs with the flexibility needed to undergo rapid turnover cycles of polymerization and depolymerization. Indeed, hydrolysis leads to a change in the angle of the longitudinal assembly, which destabilizes lateral interactions between adjacent protofilaments. The formed microtubule structure can be stabilized by the presence of a GTP cap in the exposed β tubulin monomers.

Above a critical α-β tubulin concentration, heterodimers spontaneously assemble into microtubules in vitro, resulting in MTs having different diameters because varying numbers of adjacent protofilaments form them. Conversely, microtubule nucleation in vivo is initiated from a ring-like template of γ tubulin, which guides the polymerization of MTs, involving exactly 13 proto-filaments by polymerizing from as many γ tubulin units. The nucleation process likely occurs through the recruitment of α-β dimers via α–γ interactions, which are regulated by GTP similarly to the regulation of α-β polymerization [2].

### 1.2. Microtubules as Targets for Chemotherapy

Among their various functions, perhaps the most critical role of microtubules is the formation of the mitotic spindle, as they comprise the most abundant components of the mitotic apparatus. These properties are at the core of why MTs are important targets for chemotherapy and why targeting them leads to significant side effects. Given their essential role in cell duplication, both microtubules and free tubulin have long been targeted by chemotherapeutic antimitotic agents, such as vinca alkaloids and taxanes, which act by disrupting normal MT dynamics, as any failure in microtubule assembly during spindle formation or the subsequent chromosome segregation phase typically leads to mitotic arrest and cell death. These compounds are effective anticancer agents, as they tend to impact dividing cells more than nondividing ones. However, as the structural components of microtubules, α and β tubulin are abundant proteins and make up approximately 2% of a cell’s total protein content [3], so these agents affect both cancerous and normal cells by binding tubulin indiscriminately and often lead to severe side effects. These detrimental side effects may be reduced by increasing a drug’s specificity for tubulin within cancerous cells only.

### 1.3. γ Tubulin as a Potential Alternative Target for Glioblastoma Multiforme

Glioblastoma multiforme (GBM) is among the most aggressive forms of brain tumors. The prognosis for GBM patients remains extremely poor despite advances in therapies, as they have proven unable to prolong patient survival more than a few months. The World Health Organization classification of GBM is a necrosis-predisposed grade-IV cancer that is mitotically active, which means its cells proliferate at a higher rate than normal tissue cells. During mitosis prophase, a cell prepares to divide by condensing its chromosomes and starting the formation of the mitotic spindle, of which microtubules are the main component. In vivo nucleation of microtubules begins with a complex called γ-TuRC (γ tubulin ring complex) formed by accessory proteins recalled by γ tubulin, which is, therefore, essential for the formation of microtubules. As γ tubulin is normally much less abundant than α and β tubulin, its overexpression in GBM and some breast lesions makes it an attractive target for pharmacological inhibition.

Two main isoforms of human γ tubulin exist, namely γI, encoded by gene TUBG1, and γII, encoded by gene TUBG2. They share over 97% sequence identity, so any pharmacological agent targeting one isoform would likely target both [3]. Mouse studies by Yuba-Kubo et al. showed that mouse γ-tubulin-expressing genes were orthologs of human TUBG1 and TUBG2, and it emerged from knockout experiments that γI, as opposed to γII, was essential to proper mitotic division [4]. In addition to this, studies by Katsetos et al. showed that both TUBG1 and TUBG2 were overexpressed in glioblastoma [5,6]; therefore, the analysis of the binding sites focused entirely on tubulin γI.

Katsetos et al. [7] also found that tubulin βIII was overexpressed and formed complexes with γ tubulin in glioblastomas and suggested that aberrant expression and interactions of tubulins γ and βIII may be linked to malignant changes in glial cells.

Moreover, a differential gene expression study by Wang and Zhang found the isotypes βIIa and IVa to be differentially expressed in glioblastoma compared to healthy brain tissue. They also analyzed the protein–protein interactions of the differentially expressed genes products and identified a protein interaction network made of thirty proteins, among which tubulin βIIa and βIVa were predicted to be relatively important [8].

### 1.4. Development of Gatastatin and Its Derivatives

Based on the structural similarity between γ and β tubulin, Chinen et al. [9] screened a library of β tubulin colchicine site-binding compounds, namely colchicine itself, nocodazole, plinabulin, and glaziovianin A (AG1), in order to assess whether some of them were able to bind γ tubulin as well. AG1 was found to bind to both γ and β tubulin with similar affinities, so it was derivatized to look for a selective γ tubulin binder. The findings reported in their paper showed that glaziovianin A (AG1) appeared to have similar binding affinities to α-β and γ tubulin, while its derivative, gatastatin, whose structure is shown in Figure 2, reportedly had an almost 12-fold higher affinity for γ tubulin. Moreover, the authors suggested that gatastatin’s mode of action was blocking GTP binding to γ tubulin, thus inhibiting its microtubule nucleation activity [9].

A previous study by Friesen et al. [11] already investigated the binding of colchicine site binders to γ tubulin and, conversely to the results of Chinen et al. [9], they found that colchicine seemed to bind with equal affinity to α-β and γ tubulin. Based on the high structural similarity (75%) between β and γ tubulin, they also identified a homologous region on γ tubulin corresponding to the binding site of colchicine on β tubulin.

Shintani et al. [12] performed further derivatization of gatastatin that resulted in thirteen compounds, as shown in Figure 3. Among them, O^6^-modified derivatives reportedly showed higher cytotoxicity when acting on HeLa cells than gatastatin itself. O^6^-propargyl gatastatin (**S9**), in particular, showed the lowest IC_50_ value and also seemed to be a more potent inhibitor of the GTP–γ tubulin bond in vitro. Therefore, the authors named this compound gatastatin G2 (G2).

In the present work, structural models of human α-β and γ tubulin are used to perform molecular-docking simulations and MM/GBSA calculations that provide insights on the binding sites and modes of gatastatin and its derivatives and suggest that off-target interactions may still be a potential problem. MM/GBSA is utilized to post-process docking results, as it offers a more realistic portrayal of the ligand–target binding problem when compared to docking [13]. This is mainly due to the fact that MM/GBSA takes into account the effects of solvation and entropy, which have a notable influence on the accuracy of the outcomes. Despite its computational efficiency, a trade-off between computational efficiency and accuracy is inevitable with MM/GBSA. Specifically, approximations for entropic contributions can make the method vulnerable to inaccuracies that are dependent on the system being studied [13,14]. In this study, these limitations are taken into consideration, and the resulting computational data are comparatively analyzed in a relative sense.

A microscale thermophoresis assay comparing colchicine and gatastatin binding to α-β tubulin is also performed.

Furthermore, an analysis of the compounds’ pharmacokinetic properties made with ADMET predictors highlights some potential delivery and toxicity problems.

## 2. Materials and Methods

After obtaining a library of human tubulin models, molecular-docking simulations were performed to obtain an initial pose for each ligand. Gatastatin and its thirteen derivatives, along with AG1, colchicine, KPU-406, nocodazole, and plinabulin, were docked in the GTP and in the putative colchicine-binding sites on γ tubulin, as well as in the colchicine-binding sites of α-β dimers. The target–ligand complexes were then simulated in explicit solvent, and MM/GBSA calculations were carried out. The adopted workflow is illustrated in Figure 4.

### 2.1. Modeling of Human Tubulin Isotypes

Since gatastatin is a derivative of a colchicine-site-binding compound, the search for templates of α-β tubulin for consensus docking simulations was directed toward colchicine-bound tubulin complexes. Many such structures have been deposited in the Protein Data Bank over the last two decades; the ones that were taken into consideration are listed in Table 1. The structures were compared according to these three criteria, listed in order of their importance:Resolution: any resolution between 1.5 and 2.5 Å was considered excellent.Number of missing residues (MRES): a structure with fewer missing residues was preferred. Moreover, it was best if these missing residues were not in or near known or probable active sites.Publication date: for compatibly with the other criteria, a more recent structure was preferable.

PDB entry 5EYP, representative of free tubulin, was selected, mainly for its best resolution. The structure was prepared using Molecular Operating Environment (MOE) version 2020.09 [1] to correct residues with alternate locations, missing backbone atoms in the protein termini, missing residues inside the chain, and inconsistencies between the residue name and its structure. Ionization states and position-optimized hydrogens were assigned, and energy minimization was performed to relieve strains in the structure.

The target sequences of human tubulin were downloaded as FASTA files from the UniProt KnowledgeBased (UniProtKB) database [16]; their UniProt reference IDs are reported in Table 2. 

Homology models were then obtained for each αIa-β isotype dimer in MOE using the co-complexed GTP and GDP molecules and the magnesium ions as an environment to prevent clashes and superpositions. The Protonate3D application in MOE was used to assign ionization states and position-optimized hydrogens. Upon refinement and energy minimization, the obtained models were inspected using ERRAT, Verify3D, WHATCHECK, PROCHECK, and QMEAN [17,18,19,20,21]. Particular attention was given to the overall quality scores and to the residues’ distribution on a Ramachandran plot.

As for γ tubulin, the PDB entries reported in Table 3 were inspected in terms of resolution, number of missing residues, and publication date.

PDB entry 3CB2 was selected for its best characteristics in terms of resolution and number of missing residues. The original PDB entry contained two γ chains; the A chain was retained because it had fewer missing residues than the B chain. Only preparation and minimization of the structure in MOE were necessary because 3CB2 represented a human γ tubulin structure. PDB entry 6V5V was also taken into consideration to compare the folding of the selected template, since it was the structure of the protein in the native ring complex. Their structures were superposed and inspected visually, as well as in terms of the RMSD values in MOE.

### 2.2. ADMET Analysis

The physicochemical descriptors, pharmacokinetic properties, and drug-likeness of gatastatin and its derivatives were predicted using ADMET Predictor^®^ 10.2 commercial software by Simulations Plus [22] and the online resources of SwissADME [23] and pkCSM [24]. Where applicable, the results from the three tools were then compared. Simulations Plus ADMET Predictor^®^ 10.2 is a state-of-the-art ADMET property prediction software, so, given its reliability, it was regarded as a golden standard and used as a means of comparison for the other two tools.

### 2.3. Binding Energy Predictions

#### 2.3.1. Molecular Docking

The DockBox package [25] was used to perform consensus docking simulations for γ tubulin. The AMDock suite was used to identify the optimal center and size of the docking box for each ligand [26,27]. The box was centered on the co-complexed GDP molecule to define the GTP-binding site and on the residues identified by Friesen et al. [11] to obtain the putative colchicine site. Since the upper limit of the docking box sizes suggested by AMDock was 28Å, a cubic box of size 30Å was adopted for every ligand. Docking results from MOE, AutoDock4, and AutoDock Vina were compared for consensus and rescored using DSX [28,29,30,31].

Docking in the colchicine site of α-β was performed using MOE in order to maintain the co-complexed GTP molecule on the α subunit and prevent clashes in the initial steps of the subsequent dynamics simulations. The binding site was identified based on the position of colchicine in the template structure from the PDB.

Colchicine and GDP were redocked in their binding sites on α-β and γ tubulin in the 5EYP and 3CB2 structures, respectively. The obtained poses were compared with the co-crystallized ligands in terms of RMSD.

#### 2.3.2. Molecular Dynamics

Molecular dynamics simulations were carried out using the Amber20 package [32]. The simulations were performed for each target–ligand pair as follows:Minimization with restraints;Full-structure minimization;Heating to 298 K in an NVT ensemble with a 2.0 picosecond coupling constant τ;Equilibration to 1.0 bar in an NPT ensemble with a Berendsen barostat [33], isotropic position scaling, and a coupling constant of 2.0 picoseconds;Five 2 ns production runs with different initial conditions for a total of 10 ns.

As evaluated by Sun et al., when using various simulation protocols on the PDBbind dataset, excessively long simulations could negatively affect the outcome of the MM/GBSA calculations [14]. According to this study and the common practice of calculating binding energies using MM/GBSA, we opted to perform five 2 ns simulations for a total of 10 ns for each target–ligand complex. A similar approach was successfully implemented in previous studies involving tubulin [34]. RMSD plots of the production runs for the representative α-βIII dimer and γ tubulin are provided in Figure A3 and Figure A4 in Appendix B. As the RMSD values in the second half of the simulations showed variations of the order of 1 Å or less, they were considered suitable to proceed with the binding energy calculations.

The complexes were simulated in an octahedral box using an ff14SB forcefield with explicit TIP3P water. An ionic concentration of 0.15 M was simulated by adding appropriate amounts of chlorine and sodium ions. Periodic boundary conditions were imposed on the system during the calculation of nonbonded interactions. Particle Mesh Ewald was used for long-range interaction calculations with a 10 Å cutoff.

#### 2.3.3. MM/GBSA Binding Energy Estimation

To obtain a quantitative estimation of the interactions, MM/GBSA calculations were performed using the MMPBSA.py package in Amber20.

The MM/GBSA method combines molecular mechanics (MMs) with continuum solvation models (generalized born, GB). The binding free energy is calculated as:(1)$$\Delta G_{bind}={\bar{G}}_{complex}-\left[{\bar{G}}_{protein}+{\bar{G}}_{ligand}\right],$$
where $\bar{G}$ indicates average free energy, calculated as follows:(2)$$\bar{G}={\bar{E}}_{MM}+{\bar{E}}_{GB}+{\bar{E}}_{SASA}-T{\bar{S}}_{solute}.$$

${\bar{E}}_{MM}$ is the mechanical energy in the gas phase; it is made of electrostatic, van der Waals, and internal energies (i.e., bond, angle, and dihedrals). ${\bar{E}}_{GB}$ represents the polar contribution to the solvation free energy based on the generalized born implicit solvent model, while ${\bar{E}}_{SASA}$ is the nonpolar solvation term based on the solvent-accessible surface area (SASA). Lastly, T is the absolute temperature, and ${\bar{S}}_{solute}$ is the entropy of the solute.

For each target–ligand pair, the last nanosecond of the five simulations was considered for the binding energy calculation. The results were then averaged to obtain a final estimate. 

### 2.4. Analysis

For the sake of comparison with experimental data, human β tubulin isotype sequences were compared with porcine tubulin (UniProtKB accession code P02554) to inspect their differences in the colchicine-binding site. Only the β subunit was considered because α tubulin makes little contact with colchicine compared to β. The Clustal Omega application was used to perform multiple alignment [35]. 

The estimated binding energies for α-β tubulin were averaged using the Boltzmann distribution and weighted by the relative quantities of β isotypes in a healthy human brain [36] and in an untreated glioblastoma multiforme cell line [37]. Cell line expression levels were obtained with the NCI-60 Analysis Tool in CellMiner [38]. The tubulin abundance data used for the weighted averages are reported in Table 4.

β tubulin expression data in HeLa cells were also considered for comparison with the cytotoxicity assay results by Shintani et al. [12]. Isotypes I, II, III, and IV of β tubulin had abundances of 45%, 14%, 35%, and 6% in HeLa cells, respectively [39].

For each compound, the average binding energies were obtained using the following formula:(3)$$\langle BE\rangle =\frac{\sum_{i}\Delta G_{i}\cdot f_{i}\cdot \exp\left(-\frac{\Delta G_{i}}{RT}\right)}{\sum_{i}f_{i}\cdot \exp\left(-\frac{\Delta G_{i}}{RT}\right)},$$
where *i* indicates the isotype, $f_{i}$ is the relative abundance of that isotype, $\Delta G_{i}$ is the estimated binding energy, $R=8.31\frac{J}{mol\cdot K}$ is the universal gas constant, and T is the absolute temperature (298 K).

*K_d_* predictions were then obtained from the averaged values as follows:(4)$$K_{d}=e^{\frac{\langle BE\rangle }{RT}},$$
using the same values for R and T as in Equation (3).

Binding affinities were obtained from the experimental values using:(5)$$\Delta G_{exp}=\mathrm{RTln}K_{d}.$$

Computational predictions of the compounds’ binding affinities to α- β and γ tubulin were compared to each other and with experimental results. In particular, the correlation between computationally predicted binding energies and the results of tryptophan fluorescence assays and cytotoxicity assays for HeLa cells [9,12] was investigated.

### 2.5. Microscale Thermophoresis Protocol

Microscale thermophoresis analyses were carried out using a Monolith NT.115 instrument (Nano Temper Technologies, München, Germany). α-β tubulin purified from porcine brains were labeled using a Monolith NT Protein Labeling Kit RED-NHS (Nano Temper Technologies, cat# MO-L011) following the manufacturer’s protocol. All experiments were carried out at 23 °C in Monolith NT.115 Premium capillaries (Nano Temper Technologies, cat# MO-L011), with 40% LED power (fluorescence lamp intensity) and 60% microscale thermophoresis power (IR laser intensity). The assay buffer contained 80 mM of PIPES-KOH, a pH of 6.9, 2 mM of MgCl2 and 0.5 mM of EGTA, with final DMSO concentrations of 0.4% for colchicine and 3.125% for gatastatin. Three and six replicates for colchicine and gatastatin binding to labeled tubulin were performed, respectively, and data were analyzed with Monolith Affinity Analysis v2.2.6 software, exported to excel, and plotted with GraphPad Prism 7.0.

## 3. Results

### 3.1. Analysis and Modeling of Human Tubulin Isotypes

Despite animal tubulin being the cheapest and most common choice when it comes to lab experiments, it is essential to investigate human tubulin if the focus is on clinical applications. While some experimentally resolved structures of human γ tubulin exist, human α-β dimer structures were generated in silico by homology modeling. A library of human α-β tubulin structures was obtained; the models featured tubulin αIa in complex with tubulin βI, βIIa, βIIb, βIII, βIVa, βIVb, βV, βVI, and βVIII, with a total of nine structures for each of the selected templates. 

#### 3.1.1. Human–Porcine β Tubulin Comparison in the Colchicine Site

Figure 5 reports the results of the multiple sequence alignment of human tubulin isoforms. Dots indicate perfect identity, while mismatches are color-coded according to PAM250 score [40]. Conservative replacements that scored above 0.5, indicating strong similarity, are highlighted in green. Replacements with a PAM250 score of 0.5 or less, indicating weak similarity, are semiconservative and are highlighted in orange, while red-highlighted residues are not conserved.

Since the colchicine-binding site sequences of isotypes I, IIa, IIb, IVa, and IVb were identical or highly similar to porcine β tubulin, computational results for these models were averaged for comparison with experimental results.

#### 3.1.2. Validation of Homology Models

All homology models obtained from the 5EYP template received an ERRAT score around 94%, indicating that the percentage of errors in the computational models was reasonably low. All models passed the Verify3D test, i.e., at least 80% of each structure’s amino acids scored 0.2 or above in the 3D-1D profile. Lastly, all homology models had around 91% of their amino acids in the most favored (core) regions according to the analysis with PROCHECK, and none showed residues in unfavorable regions.

#### 3.1.3. γ Tubulin Structure Inspection

The RMSD value between 3CB2 and 6V5V γ tubulin equaled 1.185Å. This result indicated a good level of similarity between the structures.

Figure 6 shows these results graphically: the structures are represented according to a color map in which a deeper shade of green indicates a lower RMSD value, while yellow, orange, and red tones mean progressively higher values. Since 6V5V had more missing residues, some portions of 3CB2 that did not match any 3D structure in 6V5V are depicted in white.

### 3.2. ADMET Analysis Results

The pharmacokinetic prediction package SwissADME was used in our study of gatastatin derivatives. The results of this analysis summarized gastrointestinal absorption, BBB permeation, and P-glycoprotein interaction with the predicted values in the Boiled-EGG evaluation, as shown in Figure 7. Boiled-EGG is acronym for Brain Or IntestinaL EstimateD permeation. It is an egg-shaped classification plot in which the “yolk” represents the physicochemical space for highly probable BBB permeation and the “white” area shows passive gastrointestinal absorption. The egg is based on WLOGP for lipophilicity and TPSA for apparent polarity. Color-coding gives information about active efflux from the brain or the GI lumen, with blue for P-glycoprotein substrates and red for nonsubstrates.

The pharmacokinetic analysis indicated that neither gatastatin nor gatastatin G2 seem to be able to cross the blood–brain barrier (BBB), which would make them unsuitable candidates for a possible therapeutic application for glioblastoma multiforme. In addition, another software package for pharmacokinetic prediction, namely ADMET Predictor^®^ produced by Simulations Plus, predicted gatastatin G2 to be an hERG inhibitor, along with the derivatives of S5, S6, S10, S11, S14, and S15, which would render these compounds potentially cardiotoxic. Of all the analyzed compounds, the only one predicted to cross the BBB was S3, which also had a better toxicity profile than gatastatin G2.

### 3.3. Molecular Docking Results

The binding scores of the top poses generated by MOE for each ligand in the colchicine-binding sites of the α-β isotypes are reported in Table 5.

Figure 8 shows the binding interactions between the gatastatin and gatastatin G2 top-scoring poses and the α-βIII structure. A summary of the interactions of the other gatastatin derivatives with α-βIII is provided in Figure A1 in Appendix A. 

The top-scoring poses of colchicine resulting from redocking in the original 5EYP template differed less than 1 Å RMSD from the original co-crystallized pose, with a score of around −9.9 kcal/mol. 

Table 6 summarizes the docking scores of the best pose resulting from consensus docking for γ tubulin. 

Redocking of GDP resulted in poses differing less than 1 Å RMSD from the co-crystallized pose, with scores around −10 kcal/mol. In light of the observations made about the pharmacokinetic profiles of gatastatin G2 and the S3 derivative, Figure 9 shows the binding interactions between the top-scoring poses of these two compounds and the γ tubulin structure in the GTP-binding site, while a summary of the interactions of the other gatastatin derivatives with γ can be found in Figure A2 in Appendix A. 

### 3.4. Predicted Preference of Gatastatin G2 for the GTP Site on γ Tubulin

While gatastatin seemed to bind with similar affinity to the two sites, gatastatin G2 exhibited a stronger affinity for the GTP-binding site. A comparison of the computational results between the GTP cleft and the colchicine-binding site on γ tubulin is reported in Table 7 in terms of binding energy estimates resulting from MM/GBSA calculations.

### 3.5. Comparison with Experimental Results

#### 3.5.1. Gatastatin Binds More Weakly Than Colchicine to α-β Tubulin

According to the microscale thermophoresis analysis, colchicine and gatastatin bound to the α-β heterodimer with higher affinities than what was found in the tryptophan fluorescence assay by Chinen et al. [9]. However, their ratios were comparable in the two cases. The results of the two studies are compared in Table 8.

#### 3.5.2. Computational Results Correlate Well with Experimental Results for γ Tubulin in the GTP-Binding Site

The binding free energies for α-β and γ tubulin are shown in Figure 10. K_d_ values resulting from the tryptophan fluorescence assay were used to obtain an estimate of the experimental binding affinity (ΔG_exp_) using Equation (3).

The correlation between the IC_50_ values resulting from the cytotoxicity assay on HeLa cells [12] and the computationally predicted binding energies on γ and α-β tubulin is shown in Figure 11. 

The experimental results showed a strong correlation with the computational results for the GTP–binding site of γ tubulin, whereas the correlation with the computational results for the colchicine-binding site of α-β tubulin was weak.

#### 3.5.3. Gatastatin Does Not Show a Clear Preference for γ Tubulin

The computationally predicted binding energies for gatastatin and its derivatives are compared in Figure 12. The MM/GBSA results for α-β tubulin were averaged and weighted based on their relative abundance in a healthy human brain and a glioblastoma multiforme cell line.

It is noticeable that the estimates of the compounds’ affinities for α-β tubulin were consistently higher than those for γ.

#### 3.5.4. Gatastatin G2 Binds More Weakly Than GTP to γ Tubulin

The binding affinity of gatastatin G2, which showed a higher affinity for the GTP-binding site than gatastatin, was compared to those of GTP and GDP. The predicted binding energy of gatastatin G2 was found to be −42.2 ± 4.9 kcal/mol, while GDP and GTP had predicted affinities of −51.7 ± 2.4 and −53.3 ± 3.1 kcal/mol, respectively.

## 4. Discussion

No indication of the specificity of the analyzed compounds for γ tubulin emerged from the comparison of gatastatin binding to γ and α-β, as shown in Figure 12. The estimates of the compounds’ affinities for α-β tubulin were consistently higher than for γ. This is in contrast with the results by Chinen et al. [9], which predicted that gatastatin bound more strongly to γ tubulin. In particular, they reported a K_d_ value of 42.5 ± 36.7 µM for the binding of gatastatin to α-β tubulin and 3.6 ± 1.3 µM for γ tubulin. It should be noted that the standard deviations of these results were 86% and 36%, respectively. Taking them into account, the lowest possible K_d_ value on α-β tubulin would be 5.8 µM, and the highest for γ tubulin would be 4.9 µM. Therefore, the claimed preference for γ tubulin may be insignificant.

In addition to this, it should be noted that α-β tubulin expression is constitutive to all cells, making up approximately 2.5% of a cell’s total protein content, while γ makes up less than 1% of the total tubulin content of a cell. Even when it is overexpressed, as it has been found for glioblastoma multiforme cancer cells, a substantially higher specificity for γ tubulin of the analyzed compounds would be needed in order to result in potent targeting in vivo. This is all the more true when considering that the proposed inhibitory mode of action of gatastatin is to prevent GTP binding to γ tubulin. The most likely way it could succeed in doing so within tolerable doses is by having a higher binding affinity to γ tubulin than GTP itself. However, this hypothesis was not supported by the computational results. Moreover, with the structural similarity of GTP-binding sites in proteins other than tubulin, off-target interaction of such compounds could lead to undesirable side effects.

## 5. Conclusions

There are numerous tubulin-targeting compounds that are effective anticancer agents, as they tend to impact dividing cells more potently than nondividing ones. However, targeting abundant α and β tubulin proteins almost always leads to severe side effects. These deleterious side effects may be reduced by increasing drugs’ specificity for tubulin isoforms, which are expressed more abundantly in cancerous cells than in healthy ones. For some tumors, a possible way to discriminate between healthy and cancerous cells may be to target γ tubulin instead of the α-β dimer since γ tubulin plays an essential role in mitosis forming nucleating rings for microtubules, particularly mitotic spindle microtubules. Thus, inhibiting its functions could have a powerful antimitotic effect. Importantly, γ tubulin has been found to be overexpressed in glioblastoma multiforme, some breast lesions, and carcinomas, while in healthy tissue cells, it is less abundant than α-β, making up less than 1% of the total tubulin content of a cell. Therefore, a compound having a strong specificity for γ tubulin could have fewer off-target interactions. The search for such compounds has been on-going for almost two decades.

For the above reasons, it was exciting news when Chinen et al. identified a potential specific and selective γ tubulin inhibitor. The results of their drug-binding assays for colchicine, AG1, and gatastatin indicated that gatastatin bound to γ tubulin with a 12-fold higher affinity than to α-β tubulin. Chinen et al. then suggested that gatastatin’s mode of action was blocking GTP binding to γ tubulin, thus inhibiting its microtubule nucleation activity. This encouraging result has not been confirmed by an in silico analysis or verified experimentally by other groups. Therefore, we undertook not only finding a molecular-level elucidation of the binding mode of gatastatin but also comparing its predicted binding affinity values for both α-β and γ tubulin.

Specifically, the present work was aimed at identifying the binding sites and binding modes of gatastatin, a potential γ tubulin inhibitor, and its recently developed derivatives. To date, other than the published results on gatastatin, there has been no success in finding a sufficiently specific and selective small molecule agent targeting γ tubulin preferentially over α-β tubulin. This is due a high level of structural similarity between these important and highly expressed proteins. Based on a combination of in silico analysis involving gatastatin and its derivatives with respect to their binding affinities for human tubulin isotypes, we concluded that, unfortunately, none of these compounds appeared likely to have a preference for γ tubulin compared to the α-β tubulin dimer. Our computational work was supported by thermophoresis assays. Consequently, we suggest further derivatization of the gatastatin scaffold guided by the computational binding predictions reported in this paper in order to search for a selective and specific γ tubulin inhibitor as a potential drug candidate for the treatment of glioblastoma multiforme. We believe that our paper opens the possibility for the rational design of a long-sought candidate drug with the desired specificity and selectivity for γ tubulin. This offers hope for finding a cure for brain cancer in the not-too-distant future.

## Figures and Tables

**Figure 1 cancers-15-01714-f001:**
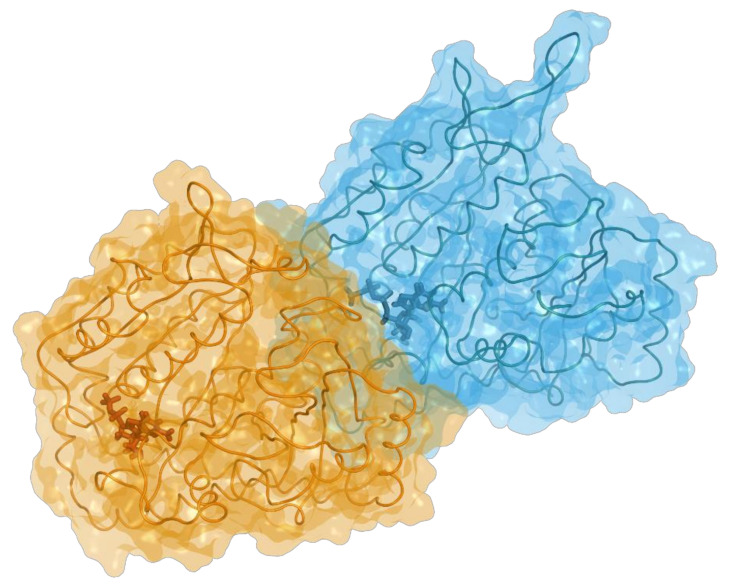
Structure of the tubulin heterodimer in complex with GTP (blue) at the nonexchangeable site and GDP (orange) at the exchangeable site on β. Figure generated using MOE v2022.02 software package [1].

**Figure 2 cancers-15-01714-f002:**
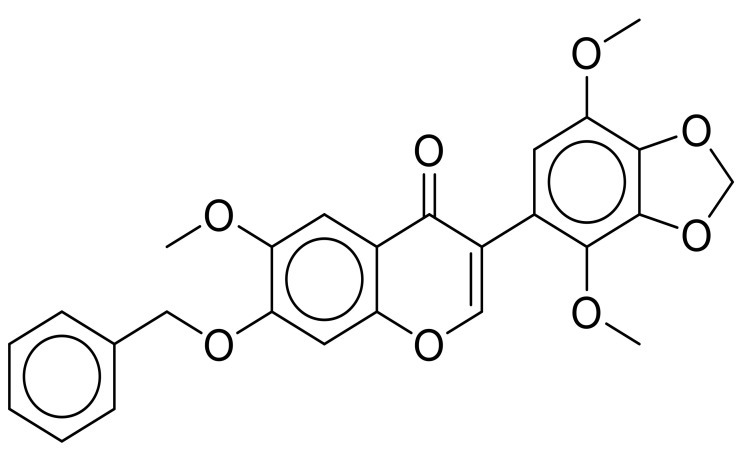
Chemical structure of the compound gatastatin. Figure obtained with ChemDraw JS 19.0.0 [10].

**Figure 3 cancers-15-01714-f003:**
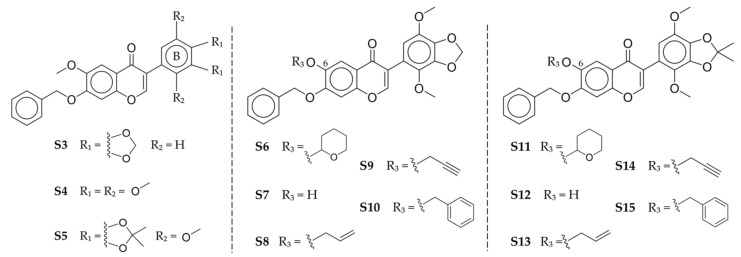
Structures of B-ring-modified (**S3**–**S5**), O^6^-modified (**S6**–**S10**), and O^6^-modified, 3′,4′ acetonide (**S11**–**S15**) gatastatin derivatives [12]. Graphics obtained with ChemDraw JS 19.0.0 [10].

**Figure 4 cancers-15-01714-f004:**
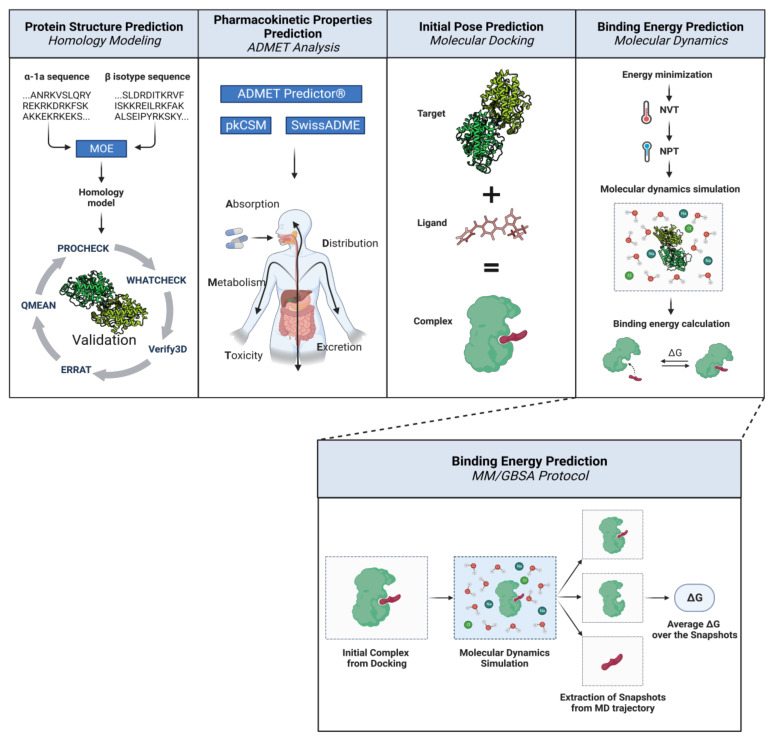
General scheme of the adopted methods for α-β dimers. First, a library of human tubulin models was obtained through homology modeling. The ligands were investigated in terms of their pharmacokinetic properties via ADMET analysis, and then their initial poses in the colchicine-binding sites were obtained through molecular-docking simulations. Lastly, a prediction of the binding energy was obtained for each target–ligand pair by running molecular dynamics simulations of the complexes, followed by MM/GBSA calculations. The MM/GBSA protocol involved extracting snapshots from the trajectory of the MD simulation, estimating the free energy of binding for each snapshot, and averaging it over all the extracted snapshots to obtain a final prediction. Illustration adapted from “Computational Biology Workflow for the Study of Protein-Protein Complexes” by BioRender.com (2023) [15]. Retrieved from https://app.biorender.com/biorender-templates, accessed on 25 November 2022.

**Figure 5 cancers-15-01714-f005:**
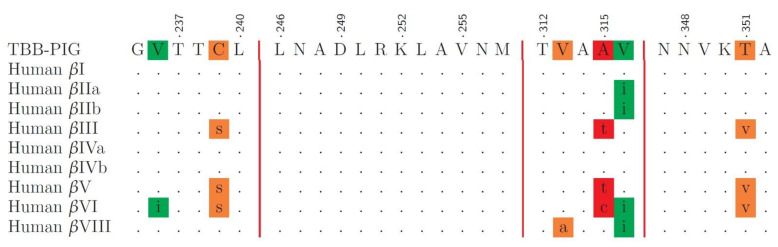
Multiple sequence alignment of porcine β tubulin and human tubulin isotypes over the residues forming the colchicine-binding site. Green-highlighted residues indicate conservative replacements, orange are semiconservative, and red-highlighted residues are not conserved.

**Figure 6 cancers-15-01714-f006:**
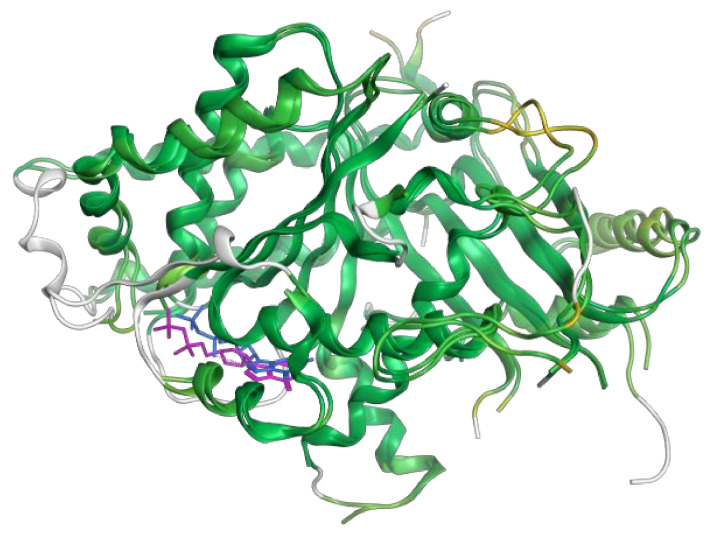
The structures of γ tubulin from PDB 3CB2 and 6V5V entries are superimposed and color-coded according to RMSD: a deeper shade of green indicates a lower RMSD value.

**Figure 7 cancers-15-01714-f007:**
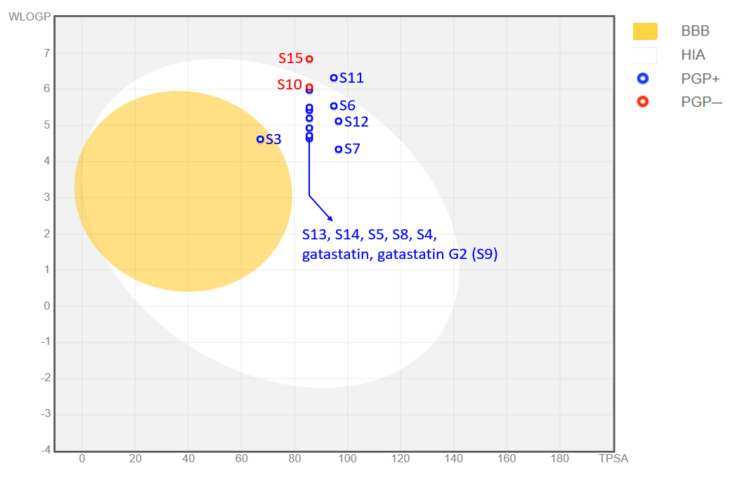
SwissADME Boiled–EGG summarizing results for gastrointestinal absorption, BBB permeation, and P-glycoprotein interactions.

**Figure 8 cancers-15-01714-f008:**
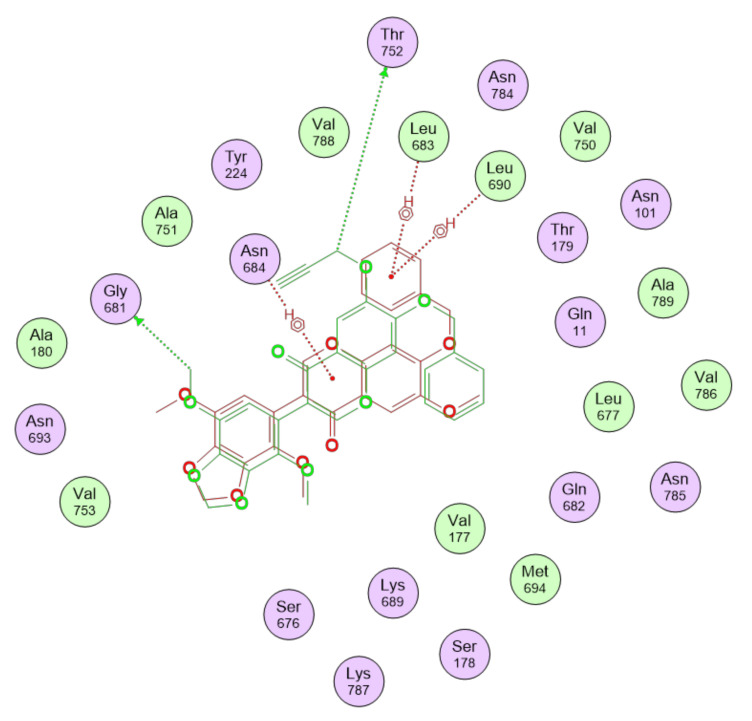
Binding interactions of gatastatin, represented in red, and gatastatin G2, shown in green, with the α-βIII isotype structural model.

**Figure 9 cancers-15-01714-f009:**
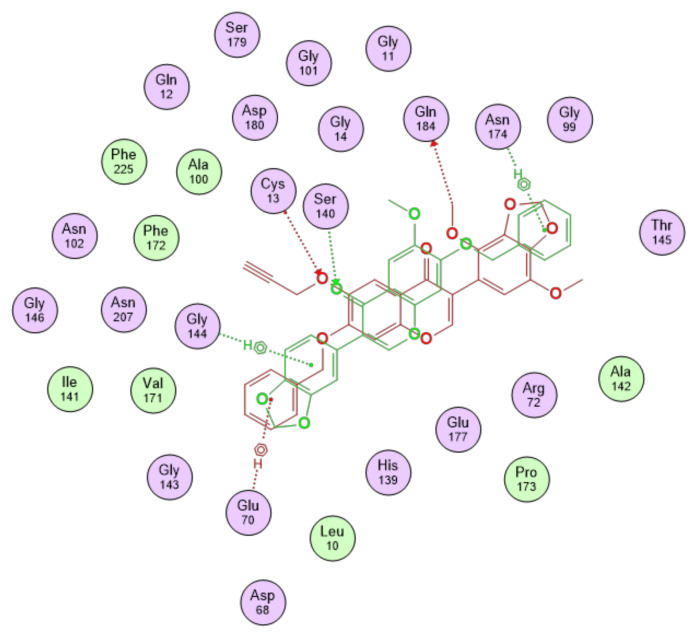
Binding interactions of gatastatin G2, represented in red, and S3 derivative, shown in green, with γ tubulin in the GTP-binding site.

**Figure 10 cancers-15-01714-f010:**
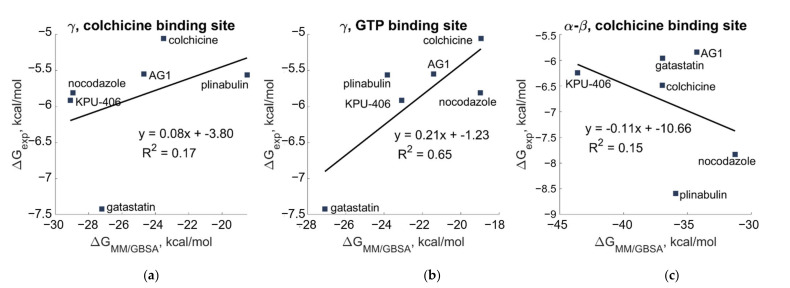
Scatterplot of binding free energy estimated from tryptophan fluorescence assay versus those calculated by MM/GBSA (**a**) for the putative colchicine-binding site on γ, (**b**) for the GTP-binding site on γ, and (**c**) for the colchicine-binding site of α-β tubulin.

**Figure 11 cancers-15-01714-f011:**
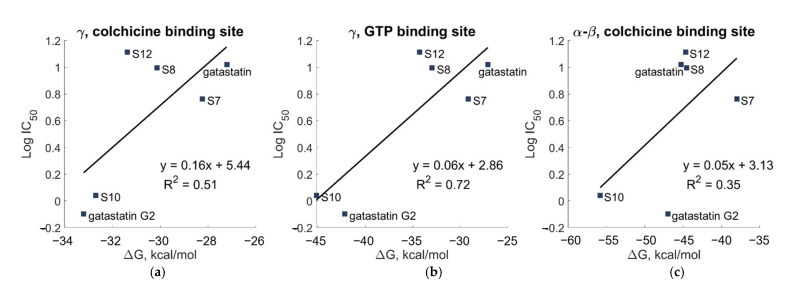
Scatterplot of experimental IC_50_ values versus the binding free energy calculated by MM/GBSA (**a**) for the putative colchicine-binding site on γ, (**b**) for the GTP-binding site on γ, and (**c**) for the colchicine-binding site of α-β tubulin.

**Figure 12 cancers-15-01714-f012:**
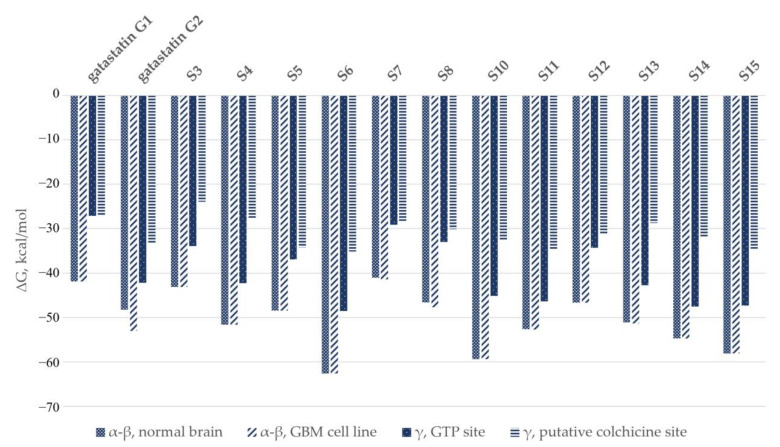
Comparison of computational results in the colchicine–binding site of α-β tubulin and the putative colchicine site and GTP site on γ tubulin.

**Table 1 cancers-15-01714-t001:** Template comparison for α-β tubulin.

PDB ID	Resolution (Å)	Number of MRES	Publication Date
1SA0	3.58	24/29 (chains A/C), 26 (chains B/D)	2004
4O2B	2.30	12 (chains A/C), 24 (chains B/D)	2014
5EYP	1.90	23 (chain A), 20 (chain B)	2016
5NM5	2.05	29 (chain A), 28 (chain B)	2017

**Table 2 cancers-15-01714-t002:** UniProtKB references of the human tubulin target sequences for homology modeling.

Isotype	UniProtKB Ref.
αIa	Q71U36
βI	P07437
βIIa	Q13885
βIIb	Q9BVA1
βIII	Q13509
βIVa	P04350
βIVb	P68371
βV	Q9BUF5
βVI	Q9H4B7
βVIII	Q3ZCM7

**Table 3 cancers-15-01714-t003:** Template comparison for γ tubulin.

PDB ID	Resolution (Å)	Number of MRES	Publication Date
1Z5V	2.71	39	2005
1Z5W	3	42	2005
3CB2	2.3	19 (chain A)	2008
6V5V	3.8	82	2020

**Table 4 cancers-15-01714-t004:** Relative quantities of β tubulin isotypes in healthy human brain and glioblastoma multiforme cell line.

Isotype	Human Brain (%)	GBM Cell Line (%)
βI	4	10
βIIa	30	19
βIIb	7	17
βIII	4	12
βIVa	46	10
βIVb	9	17
βV	0	12
βVI	0	1
βVIII	0	2

**Table 5 cancers-15-01714-t005:** Binding scores of the analyzed ligands resulting from docking simulations in the colchicine binding sites of α-β tubulin.

Compound	Score (kcal/mol)								
	α-βI	α-βIIa	α-βIIb	α-βIII	α-βIVa	α-βIVb	α-βV	α-βVI	α-βVIII
AG1	−8.34	−8.07	−8.26	−7.97	−8.34	−8.55	−8.41	−8.09	−8.42
colchicine	−9.45	−9.12	−9.01	−9.28	−9.11	−9.61	−8.75	−9.81	−8.91
KPU−406	−9.62	−9.51	−9.42	−9.52	−9.38	−9.99	−0.97	−9.29	−9.38
nocodazole	−7.43	−7.09	−7.19	−7.06	−7.46	−7.31	−7.31	−7.34	−7.46
plinabulin	−8.27	−8.06	−7.80	−8.13	−7.82	−8.35	−7.96	−8.01	−8.01
gatastatin G1	−9.79	−9.55	−9.50	−9.48	−9.15	−9.63	−9.49	−9.51	−9.68
gatastatin G2	−10.10	−10.17	−9.95	−9.92	−9.79	−10.00	−10.10	−10.17	−9.99
S3	−8.63	−8.63	−8.73	−8.65	−8.55	−8.58	−8.59	−8.60	−8.76
S4	−9.87	−10.16	−9.78	−9.84	−9.88	−9.60	−9.61	−9.19	−9.55
S5	−9.74	−9.22	−9.88	−9.69	−9.66	−9.69	−8.99	−9.42	−8.97
S6	−10.28	−10.48	−9.95	−9.60	−10.32	−10.65	−1.01	−10.55	−10.91
S7	−8.66	−9.38	−9.29	−9.09	−9.04	−9.38	−9.07	−9.01	−8.96
S8	−10.24	−9.74	−10.01	−9.71	−10.25	−9.76	−9.99	−10.03	−10.07
S10	−9.98	−10.79	−10.83	−10.10	−10.54	−10.51	−10.55	−10.31	−10.34
S11	−10.13	−10.61	−10.13	−9.50	−10.16	−10.38	−10.17	−9.73	−10.03
S12	−9.08	−9.05	−9.24	−9.12	−9.20	−9.28	−8.97	−9.29	−8.95
S13	−9.72	−9.54	−9.91	−9.93	−10.12	−9.76	−10.36	−10.29	−9.78
S14	−9.72	−10.22	−10.14	−9.96	−9.91	−9.45	−9.73	−10.24	−10.09
S15	−10.35	−10.73	−10.70	−10.42	−10.64	−11.09	−10.10	−9.93	−10.84

**Table 6 cancers-15-01714-t006:** Binding scores of the analyzed ligands resulting from docking simulations in the GTP site and in the putative colchicine site of γ tubulin.

Compound	Score (kcal/mol)	
	GTP Site	Putative Colchicine Site
AG1	−6.6	−6.33
colchicine	−7.34	−6.57
KPU-406	−8.53	−8.72
nocodazole	−6.15	−6.50
plinabulin	−7.78	−7.43
gatastatin G1	−7.86	−7.94
gatastatin G2	−9.06	−8.29
S3	−8.71	−7.27
S4	−8.01	−7.59
S5	−10.38	−7.56
S6	−10.41	−9.03
S7	−8.50	−8.08
S8	−7.92	−7.75
S10	−10.39	−9.44
S11	−10.25	−9.20
S12	−10.53	−8.86
S13	−10.93	−7.99
S14	−9.60	−7.40
S15	−9.94	−7.80

**Table 7 cancers-15-01714-t007:** Comparison of predicted binding affinities of gatastatin and gatastatin G2 in the GTP site and in the putative colchicine-binding site on γ tubulin.

	ΔG (kcal/mol)		
	GTP-Binding Site	Colchicine Putative Site	Ratio
Gatastatin	−27.1 ± 3.7	−27.2 ± 2.1	0.99
Gatastatin G2	−42.2 ± 4.9	−33.2 ± 4.3	1.27

**Table 8 cancers-15-01714-t008:** Comparison of experimentally derived and computationally predicted K_d_ values (µM) for gatastatin and colchicine binding to α-β tubulin.

	Chinen et al.	Microscale Thermophoresis
Gatastatin	42.5 ± 36.7	2.16 ± 0.47
Colchicine	17.5 ± 2.7	0.676 ± 0.11
Ratio	2.4	3.2

## Data Availability

All data that were generated or analyzed during this study are available from the corresponding authors upon justified request.

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
