# Peer review of "Computational Analysis and Experimental Testing of the Molecular Mode of Action of Gatastatin and Its Derivatives"

_cancers, 2023, doi:10.3390/cancers15061714_

Round 1
Reviewer 1 Report
This is a computational chemistry study performed to test if a literature compound that is claimed to be a specific inhibitor of gamma tubulin which competes at the GTP binding site is likely to be specific. The results are convincingly negative. Predicted binding of the compound is higher to beta tubulin. Also, the compound is predicted to not compete well with GTP at the GTP site. The work is significant because it casts strong doubt on use of gatastatin as a tool to study the function of gamma tubulin. There are no other tool compounds for this target. The work is limited because it is computational with no experimental validation. Still, it will hopefully force groups who want to use gatastatin to be critical of their results
Author Response
Reviewer #1:
This is a computational chemistry study performed to test if a literature compound that is claimed to be a specific inhibitor of gamma tubulin which competes at the GTP binding site is likely to be specific. The results are convincingly negative. Predicted binding of the compound is higher to beta tubulin. Also, the compound is predicted to not compete well with GTP at the GTP site. The work is significant because it casts strong doubt on use of gatastatin as a tool to study the function of gamma tubulin. There are no other tool compounds for this target. The work is limited because it is computational with no experimental validation. Still, it will hopefully force groups who want to use gatastatin to be critical of their results.
Answer: We are pleased to learn that the reviewer found our manuscript to be acceptable without requiring any changes. We are very appreciative of the reviewer for their careful reading of our manuscript and for providing valuable feedback.
Reviewer 2 Report
In this work, the authors performed molecular docking, molecular dynamics simulations and MMGBSA calculations on gatastatin and its derivatives and compared predicted binding free energies with experimental microscale thermophoresis assay results.
I hope the authors can address my comments below:
-
Figure 4 (Binding energy calculation): Since the authors used MMGBSA to calculate the binding free energies, this figure does not accurately illustrate how MMGBSA works. In fact, there is no “binding” in MMGBSA calculation and the calculation is purely done on conformations of the bound state (which is one of the reasons why MMGBSA is not accurate in many cases).
-
Line 215: what is the force field used for protein and ligand in simulations?
-
Line 222: For a protein-ligand complex system with a size like the one in this work, 2 ns MD simulation is not enough for sufficient sampling. Even though 5 independent runs were performed the timescale is still not long enough. I suggest the authors either run longer simulations (at least 100 ns for each run) or provide some evidence to show that the current simulation timescale indeed is enough to provide converged and reliable results. This is very important since the MMGBSA calculation is performed based on the sampled conformations in MD simulations. So this also affects the predicted binding free energies which is a key focus of this work.
-
MMGBSA is known as a cheap but less reliable method. It has been suggested by some work that this method can indeed provide a better estimate of binding potency rank for small molecules but still cannot achieve an accuracy as more robust methods such as Free Energy Perturbation or Thermodynamic Integration. (see a recent paper on this topic: https://doi.org/10.1002/cmdc.202200425). It also has been suggested by other papers that MMGBSA is not accurate at all (see: https://doi.org/10.1007/s10822-019-00240-w). Some papers also suggest MMGBSA is accurate in some cases (https://pubs.acs.org/doi/pdf/10.1021/ci100275a). So the accuracy of MMGBSA is system dependent and is also sensitive to parameters used in this calculation. I suggest the authors perform more robust calculations to predict binding free energies. If not, the authors should at least discuss these previous work done on MMGBSA to clarify the accuracy of this method (with the mentioned papers or more from literature). Otherwise it is possible to leave the readers an inaccurate impression about MMGBSA.
-
Figure 7,8 and Section 3.3: Based on the correlation coefficient, I do not think the predicted binding free energies from MMGBSA correlates well with the experimental data. I would not consider a R^2 value below 0.5 as significant to show a good correlation. In fact, I think this poor correlation is partially due to the relative short MD simulations performed on these systems. Again, MMGBSA used an inaccurate scoring function to calculate the energy which does not best represent the underlying physics. This can accelerate the calculation but the accuracy suffers. Besides, the input conformation for MMGBSA calculation is from MD simulation. So if the MD simulation is not converged, the accuracy of MMGBSA is also affected.
Author Response
Reviewer #2:
We thank the reviewer for the constructive feedback. We considered the feedback from the reviewer in detail and further improved the paper as the reviewer suggested. Our detailed responses to the comments and questions are below.
- Figure 4 (Binding energy calculation): Since the authors used MMGBSA to calculate the binding free energies, this figure does not accurately illustrate how MMGBSA works. In fact, there is no “binding” in MMGBSA calculation and the calculation is purely done on conformations of the bound state (which is one of the reasons why MMGBSA is not accurate in many cases).
Answer: Figure 4 has been modified to describe the workflow of MMGBSA calculations better. The new figure can be found in the Materials and Methods section at line 221.
- Line 215: what is the force field used for protein and ligand in simulations?
Answer: We added the following sentence to the Material and Methods section (line 311):
“The complexes were simulated in an octahedral box using the ff14SB forcefield with explicit TIP3P water.”
- Line 222: For a protein-ligand complex system with a size like the one in this work, 2 ns MD simulation is not enough for sufficient sampling. Even though 5 independent runs were performed the timescale is still not long enough. I suggest the authors either run longer simulations (at least 100 ns for each run) or provide some evidence to show that the current simulation timescale indeed is enough to provide converged and reliable results. This is very important since the MMGBSA calculation is performed based on the sampled conformations in MD simulations. So this also affects the predicted binding free energies which is a key focus of this work.
Answer: As evaluated by Sun et al. using various simulations protocols on the PDBbind dataset, excessively long simulations may negatively affect the outcome of the MM/GBSA calculations [https://pubmed.ncbi.nlm.nih.gov/24999761/]. They performed simulations of different lengths, ranging from 0.1 ns to 20 ns, and the 1 ns production runs were used to evaluate the prediction accuracy of the MM/GBSA method. According to this study and the common practice to calculate binding energies using MM/GBSA, we opted to perform five 2 ns simulations, for a total of 10 ns for each target-ligand complex. A similar approach has been successfully implemented in previous studies involving tubulin [https://doi.org/10.3390/molecules25081789].
RMSD plots of the production runs for the representative α-βIII dimer and γ tubulin are provided in Appendix B. As the RMSD in the second half of the simulations shows variations of the order of 1 Å or less, they were considered reasonably stable and suitable to proceed with the binding energy calculations.
- MMGBSA is known as a cheap but less reliable method. It has been suggested by some work that this method can indeed provide a better estimate of binding potency rank for small molecules but still cannot achieve an accuracy as more robust methods such as Free Energy Perturbation or Thermodynamic (see a recent paper on this topic: https://doi.org/10.1002/cmdc.202200425). It also has been suggested by other papers that MMGBSA is not accurate at all (see: https://doi.org/10.1007/s10822-019-00240-w). Some papers also suggest MMGBSA is accurate in some cases (https://pubs.acs.org/doi/pdf/10.1021/ci100275a). So the accuracy of MMGBSA is system dependent and is also sensitive to parameters used in this calculation. I suggest the authors perform more robust calculations to predict binding free energies. If not, the authors should at least discuss these previous work done on MMGBSA to clarify the accuracy of this method (with the mentioned papers or more from literature). Otherwise it is possible to leave the readers an inaccurate impression about MMGBSA.
Answer: We thank the reviewer for the suggestions. The following discussion has been added to the Introduction (line 184):
“MM/GBSA was utilized to post-process docking results as it offers a more realistic portrayal of the ligand-target binding problem when compared to docking. [http://dx.doi.org/10.1002/cmdc.202200425]. This is mainly due to the fact that MM/GBSA takes in to account the effects of solvation and entropy, which have a notable influence on the accuracy of the outcomes. Despite its computational efficiency, a trade-off between computational efficiency and accuracy is inevitable with MM/GBSA. Specifically, approximations for entropic contributions can make the method vulnerable to inaccuracies that are dependent on the system being studied [http://dx.doi.org/10.1002/cmdc.202200425, http://dx.doi.org/10.1039/c4cp01388c]. In this study, these limitations were taken into consideration, and the resulting computational data were comparatively analyzed in a relative sense.”
- Figure 7,8 and Section 3.3: Based on the correlation coefficient, I do not think the predicted binding free energies from MMGBSA correlates well with the experimental data. I would not consider a R^2 value below 0.5 as significant to show a good correlation. In fact, I think this poor correlation is partially due to the relative short MD simulations performed on these systems.
Again, MMGBSA used an inaccurate scoring function to calculate the energy which does not best represent the underlying physics. This can accelerate the calculation but the accuracy suffers. Besides, the input conformation for MMGBSA calculation is from MD simulation. So if the MD simulation is not converged, the accuracy of MMGBSA is also affected.
Answer: Our text mentions a good correlation just for gamma in the GTP binding site (line 528, “The experimental results show a strong correlation with the computational results for the GTP binding site of γ tubulin, whereas the correlation with the computational results for the colchicine binding site of α-β tubulin is weak.”)
We modified the title of section 3.5.2 in the Results section (line 509) to provide greater clarity on this matter.
Our computational analysis revealed that there was a lack of correlation between some of the experimental data points presented in Chinen et al.’s study and our computational results. These findings may raise questions about the actual specificity of gatastatin and suggest the need for further research to fully understand its mechanism of action.
Appendix B includes Figures B1 and B2, which depict the RMSD plots for the α-βIII and γ tubulin structures in complex with all of the ligands investigated in the study. The RMSD values in the second half of the simulations displayed variations of 1 Å or less, indicating that the structures were reasonably stable and suitable for conducting the binding energy calculations.
Reviewer 3 Report
Comments
1. Was the protein protonated prior to molecular docking studies? if so, inform how it was performed, if not, redo the studies by docking.
2. I suggest creating a table with the results of molecular docking studies including redocking (with the GDP ligand and the 5eyp and 3cb2 structures) and discussing these results.
2. The ADMET analysis should be represented in Figure 4, in a step before molecular docking studies.
Author Response
Reviewer #3:
We thank the reviewer for the constructive feedback. We considered the feedback from the reviewer in detail and further improved the paper as the reviewer suggested. Our detailed responses to the comments and questions are below.
- Was the protein protonated prior to molecular docking studies? If so, inform how it was performed, if not, redo the studies by docking.
Answer: The following sentence was added to the Materials and Methods section (line 247): “The Protonate3D application in MOE was used to assign ionization states and position-optimized hydrogens.”
- I suggest creating a table with the results of molecular docking studies including redocking (with the GDP ligand and the 5eyp and 3cb2 structures) and discussing these results.
Answer: We thank the reviewer for the suggestion. A dedicated section 3.3 (line 456) has been added in the Results for discussion of molecular docking.
- The ADMET analysis should be represented in Figure 4, in a step before molecular docking studies.
Answer: Figure 4 was modified according to the reviewer’s suggestion to give a better overview of the work. The updated figure can be found in the Materials and Methods section (line 221). The Materials and Methods and Results sections were also restructured to reflect the order in which the steps are presented in Figure 4.
Reviewer 4 Report
The work is original. The study IS correct
Authors are encouraged to shorten the abstract to briefly highlight the aim of the study and its main findings.
Although the introduction section conveys important info related to the aim of the study but it lacks a logical flow of the presented info, so authors are encouraged to restructure the introduction.
Binding interactions should be provided in the form of figures at least in the supplementary file.
The conclusion should be concise, as it's too long for the reader. Moreover, some parts should be shifted to the discussion part.
remove references from the conclusion part.
Although the manuscript was found interesting, it requires revision before it can be accepted for publication. Especially, triplicate validation of simulation is required as a piece of evidence confirming the binding state.
Author Response
Reviewer #4:
We thank the reviewer for the constructive feedback. We considered the feedback from the reviewer in detail and further improved the paper as the reviewer suggested. Our detailed responses to the comments and questions are below.
- Authors are encouraged to shorten the abstract to briefly highlight the aim of the study and its main findings.
Answer: We agree with the reviewer’s suggestion to shorten the abstract and will ensure that the revised abstract provides a clear and succinct summary of our study. The Simple Summary has also been restructured to describe the aim of our work better.
“Simple Summary: The glaziovianin A derivative gatastatin, presented as a γ tubulin-specific inhibitor, could represent a viable chemotherapeutic strategy to solve the specificity issues associated with targeting α and β tubulin. Since gatastatin’s specificity for γ tubulin has not been con-firmed by in silico analysis or verified experimentally by other groups, we undertook to find a molecular-level elucidation of the binding mode of gatastatin and to compare its predicted binding affinity values for both α-β and γ tubulin. We believe that our paper opens a possibility for a rational design of the long-sought candidate drug with the desired specificity and selectivity for γ tubulin.
Abstract: Given its critical role in cell mitosis, the tubulin γ chain represents a viable chemotherapeutic target to solve the specificity issues associated with targeting α and β tubulin. Since γ tubulin is overexpressed in Glioblastoma Multiforme (GBM) and some breast lesions, the glaziovianin A derivative gatastatin, presented as a γ tubulin-specific inhibitor, could yield a successful therapeutic strategy. The present work aims to identify the binding sites and modes of gatastatin and its derivatives through molecular docking simulations. Computational binding free energy predictions were compared to experimental microscale thermophoresis assay results. The computational simulations did not reveal a strong preference towards γ tubulin, suggesting that further derivatization may be needed to increase its specificity.”
- Although the introduction section conveys important info related to the aim of the study but it lacks a logical flow of the presented info, so authors are encouraged to restructure the introduction.
Answer: We thank the reviewer for the suggestion. The introductory section has been restructured into subsections in an effort to provide a clearer presentation of the information.
- Binding interactions should be provided in the form of figures at least in the supplementary file.
Answer: Figures depicting the binding interactions can be found in both the Results section (section 3.3, line 457) and Appendix A.
- The conclusion should be concise, as it's too long for the reader. Moreover, some parts should be shifted to the discussion part. Remove references from the conclusion part.
Answer: To streamline the Conclusions section, we restructured it to make it more concise and removed the references.
- Although the manuscript was found interesting, it requires revision before it can be accepted for publication. Especially, triplicate validation of simulation is required as a piece of evidence confirming the binding state.
Answer: Five 2-ns-long simulations were performed for each protein-ligand complex, and the results of MM/GBSA calculations of the five runs were averaged to obtain the final predictions
Round 2
Reviewer 2 Report
The authors have addressed my comments. Nice work!
Reviewer 4 Report
All the corrections have been made by the authors. This is suitable for publication now.